# Influence of Physicochemical Characteristics of Bean Crop Soil in *Trichoderma* spp. Development

Sara Mayo-Prieto [1,*] , Alejandra J. Porteous-Álvarez [1] , Sergio Mezquita-García [2] , Álvaro Rodríguez-González [1] , Guzmán Carro-Huerga [1] , Sara del Ser-Herrero [1] , Santiago Gutiérrez [3] and Pedro A. Casquero [1]

[1] Grupo Universitario de Investigación en Ingeniería y Agricultura Sostenible (GUIIAS), Instituto de Medio Ambiente, Recursos Naturales y Biodiversidad, Universidad de León, Avenida Portugal 41, 24071 León, Spain; apora@unileon.es (A.J.P.-Á.); alrog@unileon.es (Á.R.-G.); gcah@unileon.es (G.C.-H.); sserh@unileon.es (S.d.S.-H.); pacasl@unileon.es (P.A.C.)
[2] Department of Clinical Microbiology and Infectious Diseases, Hospital General Universitario Gregorio Marañón, Universidad Complutense de Madrid, 28007 Madrid, Spain; sergio.mezquitag@hotmail.com
[3] Grupo Universitario de Investigación en Ingeniería y Agricultura Sostenible (GUIIAS), Área de Microbiología, Escuela de Ingeniería Agraria y Forestal, Universidad de León, Campus de Ponferrada, Avenida Astorga s/n, 24400 Ponferrada, Spain; s.gutierrez@unileon.es
* Correspondence: smayp@unileon.es

**Abstract:** Spain has ranked 6th on the harvested bean area and 8th in bean production in the European Union (EU). The soils of this area have mixed silt loam and sandy loam texture, with moderate clay content, neutral or acidic pH, rich in organic matter and low carbonate levels, providing beans with high water absorption capacity and better organoleptic qualities after cooking. Similar to other crops, it is attacked by some phytopathogens. Hitherto, chemical methods have been used to control these organisms. However, with the Reform of the Community Agrarian Policy in the EU, the number of authorized plant protection products has been reduced to prevail food security, as well as to be sustainable in the long term, giving priority to the non-chemical methods that use biological agents, such as *Trichoderma*. This study aimed to investigate the relative importance of various crop soil parameters in the adaptation of *Trichoderma* spp. autoclaved soils (AS) and natural soils (NS) from the Protected Geographical Indication (PGI) "Alubia La Bañeza—León" that were inoculated with *Trichoderma velutinum* T029 and *T. harzianum* T059 and incubated in a culture chamber at 25 °C for 15 days. Their development was determined by quantitative PCR. Twelve soil samples were selected and analyzed from the productive zones of Astorga, La Bañeza, La Cabrera, Esla-Campos and Páramo. Their physicochemical characteristics were different by zone, as the texture of soils ranged between sandy loam and silt loam and the pH between strongly acid and slightly alkaline, as well as the organic matter (OM) concentration between low and remarkably high. Total C and N concentrations and their ratio were between medium and high in most of the soils and the rest of the micronutrients had an acceptable concentration except for Paramo's soil. Both *Trichoderma* species developed better in AS than in NS, *T. velutinum* T029 grew better with high levels of OM, total C, ratio C:N, P, K, Fe, and Zn than *T. harzianum* T059 in clay soils, with the highest values of cation exchange capacity (CEC), pH, Ca, Mg and Mn. These effects were validated by Canonical Correlation Analysis (CCA), texture, particularly clay concentration, OM, electrical conductivity (EC), and pH (physical parameters) and B and Cu (soil elements) are the main factors explaining the influence in the *Trichoderma* development. OM, EC, C:N ratio and Cu are the main soil characteristics that influence in *T. velutinum* T029 development and pH in the development of *T. harzianum* T059.

**Keywords:** soil physicochemical characteristics; real-time quantitative PCR (qPCR); *Trichoderma velutinum*; *Trichoderma harzianum*; bean; canonical correlation analysis

## 1. Introduction

The common bean (*Phaseolus vulgaris* L.) is one of the most important legumes crops worldwide together with the soybean (*Glycine max* (L.) Merr.) and peanut (*Arachis hypogea* L.). On a global scale the harvested area of beans was 34,495,662 ha in 2018 with a production of 30,434,280 tonnes, with the biggest producers being India, Brazil and Myanmar. In the European Union (EU), 206,076 ha and 401,609 tonnes were produced in the same year, and Spain ranked 6th on the harvested area (4.5% of the EU) and 8th in production (4.3% of the EU) behind countries such as Lithuania, Latvia or Poland [1]. Focusing on Spain, León, located in the northwest, is the main common bean producer with almost 52% of the harvested area and over 57% of the national production in 2018 [2]. The socioeconomic peculiarities of this production area have made possible the maintenance of the high quality of the common bean, which are the use of traditional varieties ("Canela, Riñón, Pinta, Plancheta"), grown in smallholding (between 1 ha and 10 ha) by the own supplying and sale in local and national markets [3]. It is for these characteristics that there is a Protected Geographical Indication (PGI), called "Alubia La Bañeza-León", (EC Reg. n. 256/2010 published on 26 March 2010, OJEU L880/17) sheltering the high quality of this product. The productive area of this PGI comprises the zones of the "Astorga", "El Páramo", "Esla-Campos", "La Bañeza", "La Cabrera", and "Tierras de León" as well as the region of "Benavente-Los Valles" in the province of Zamora, adjacent to the previous one. The soils of this area have between silt loam and sandy loam texture, with moderate clay content, neutral or acidic pH, rich in organic matter and low carbonate levels. Together, these soils provide beans with high water absorption capacity and better organoleptic qualities after cooking.

*P. vulgaris* is affected by phytopathogens like other crops. For fungal control, fungicides applied on the seed or directly to the soil can be efficient during germination or in a subsequent short period, but occasionally root rot or yellowing and wilting are not efficiently controlled [4–7]. With the Reform of the European Community Agrarian Policy (CAP), the number of authorized plant protection products has been cut down to prevail food security, as well as to be sustainable in the long term, giving priority to the non-chemical methods. Soils are physically, chemically, and biologically heterogeneous, and any variation in these characteristics might affect the development of any biocontrol agent (BCA). BCAs have different ability of growth, adaptation, or plant protection, depending on the environmental peculiarities. Nevertheless, some abiotic soil factors, such as pH, mineral fertilizers, organic matter, can modify these processes. There are some studies indicating that the composition of the environment influences microbial community composition [8–11].

The improvement of the conditions for bean establishment in the field has been investigated to get a better control of phytopathogenic fungi. For example, by developing and evaluating new sowing techniques combined with the use of pesticides for optimizing the emergence of seeds [7,12,13]. Another strategy has been the use of BCAs, reducing the dependence on synthetic chemical products. Some of the BCAs most commonly used are bacteria such as *Agrobacterium*, *Pseudomonas*, *Streptomyces,* or *Bacillus* and fungi like *Gliocladium*, *Trichoderma*, *Ampelomyces*, *Candida*, and *Coniothyrium*. *Trichoderma* spp. (Teleomorph: *Hypocrea*) is often used as a BCA for plant disease control. It exhibits a fast growth, and it is an opportunistic and non-virulent symbiont [14]. It is able to colonize the root surface, causing substantial changes in plant metabolism [15]. Furthermore, it attacks fungal phytopathogens by competing for nutrients and establishing a mycoparasitic relationship [16]. *Trichoderma* also induces the expression of genes involved in plant defence response, and besides, it promotes plant development and nutrient uptake [14,17–23].

Nevertheless, the development of biocontrol strategies requires the availability of accurate analytical tools for monitoring fungal growth in different substrates or culture soils. Real-time quantitative PCR (qPCR) is a standard method for the detection and quantification of fungal populations such as *Trichoderma* spp. [24,25] and *Gliocladium* spp. [26], but also of some soilborne pathogens such as *Rhizoctonia* spp. [27], *Fusarium* spp. [28,29], or *Pythium* spp. [30].

This study aims to investigate the influence of soil physicochemical characteristics in *Trichoderma* strains development. Thus, a protocol to select *Trichoderma* strains depending on the soil characteristics is going to be described.

## 2. Materials and Methods

### 2.1. Trichoderma *Isolates Used in this Study*

*Trichoderma velutinum* T029 and *T. harzianum* T059 were collected from the production area of the PGI "Alubia La Bañeza—León", T029 was isolated from a productive area of Astorga (Otero de Escarpizo) and T059 of Páramo (La Milla del Páramo) [31]. They were selected based on their capacity as a BCAs against phytopathogens and their positive effect on bean crops. They were stored in the collection of the Research Group of Engineering and Sustainable Agriculture (GUIIAS) from the University of León (León, Spain). Both isolates were inoculated on Petri dishes with potato-dextrose-agar (PDA, Sigma Aldrich, Germany) medium and grown at 25 °C in the dark for one week.

### 2.2. Standard Curve of Each Isolated

Fungal genomic DNA isolation was carried out as previously described by Cardoza et al. [32].

A standard curve was obtained with 320, 160, 80, 40, 20, and 10 ng DNA of every *Trichoderma* isolate, following the described previously procedure [18]. Each measurement was conducted in triplicate. Step One Plus™ (Applied Biosystems, California, USA) and *α-actin* as the reference gene for all analyses were used for the qPCR reactions according to the procedure described by Mayo-Prieto et al. [24].

### 2.3. Soil Sampling

Soil samples were collected per plot from horizon A (10 cm depth), using an unaligned sampling design within a defined area from PGI area where the bean crop was a part of the crop rotation strategy, according to a previous study [33]. The number of samples collected in each production area was proportional to the area dedicated to the bean production (Table 1).

**Table 1.** Places and code of the soil sampling used for this study.

| Productive Zone | Code | Place | Crop |
|---|---|---|---|
| Astorga | A1 | Otero de Escarpizo | *Phaseolus vulgaris* (Bean) |
| | A2 | Otero de Escarpizo | *Triticum aestivum* (Wheat) |
| | A3 | Sueros de Cepeda | *P. vulgaris* (Bean) |
| La Bañeza | B2 | San Juan de Torres | *P. vulgaris* (Bean) |
| La Cabrera | C1 | Castrocontrigo | *P. vulgaris* (Bean) |
| | C2 | Castrocontrigo | *Solanum tuberosum* (Potato) |
| Esla-Campos | E1 | Jabares de los Oteros | *P. vulgaris* (Bean) |
| Páramo | P1 | Bercianos del Páramo | *P. vulgaris* (Bean) |
| | P2 | Bercianos del Páramo | *Helianthus annus* (Sunflower) |
| | P3 | Bustillo del Páramo | *P. vulgaris* (Bean) |
| | P4 | La Milla del Páramo | *Beta vulgaris* (Beat) |
| | P5 | La Milla del Páramo | *P. vulgaris* (Bean) |

### 2.4. Physicochemical Analysis of Soils

The physicochemical analysis was carried out in the Laboratory of Instrumental Techniques of the University of León (León, Spain), according to the official methods of the Spanish Ministry of Agriculture [34].

Previously, a portion of the soils was ground with a ball mill, obtaining a smaller particle size for the determinations of carbon, nitrogen, carbonates, and organic matter.

For the determination of the textural class, it was used the Bouyoucos densimeter method and graphically represented by the triangle of the United States Department of Agriculture (USDA). Regarding pH, it was measured in a soil:water suspension in a 1:2.5 ratio and subsequently read by the potentiometric method, using a pH meter. For electrical conductivity (EC), a soil:water suspension in a 1:5 ratio was read by the conductometric method, using a conductivity meter. In the case of carbonate concentration, it was determined by Bernard's calcimeter. Organic matter (OM) concentration was determinaed by following the Walkley-Black method. For total nitrogen (N) and total carbon (C) concentrations were obtained by the Dumas method with an elemental analyzer EURO EA 3000. The Olsen method was used for the determination of assimilable phosphorus (P) by extraction with a 0.5 M $NaHCO_3$ pH 8.5 solution and reading by molecular spectrometry using a UV/VIS spectrophotometer.

Regarding the cations, as potassium (K), calcium (Ca), magnesium (Mg), and sodium (Na) were extracted by $AcONH_4$ 1N pH 7 solution and read by Inductively Coupled Plasma Optical Emission Spectrometry (ICP-OES). The cation exchange capacity (CEC) was determined using a 0.1M barium chloride solution and read by ICP-OES. As for the trace elements as iron (Fe), manganese (Mn), zinc (Zn), and copper (Cu), they were extracted with a diethylene triamine pentaacetic acid (DTPA) pH 7.3 solution and read by ICP-OES. Finally, boron (B) extraction was carried out using hot water and ICP-OES.

### 2.5. Soil Inoculation

Soil samples were divided into autoclaved soil (AS) and natural soil (NS). The AS was autoclaved at 121 °C 20 min eliminating any present microorganism. The NS went through no sterilization process.

Soil was weighed (5 g) and placed in a Petri dish of 60 mm in diameter. 5 mL of autoclaved distilled water (121 °C 20 min) were added to each one. 1 mL of a solution of $2 \times 10^7$ spores·mL$^{-1}$ of each *Trichoderma* isolate to each Petri dish. Plates were left about 2 h in a laminar flow hood to get the tilth humidity. Petri dishes were sealed with Parafilm® and incubated in a culture chamber at 25 °C for 15 days in the dark. Three repetitions of each treatment were performed. After this period, they were stored at −80 °C until processed.

### 2.6. DNA Extraction of Soil Samples and Real Time-PCR (qPCR) Analysis

DNA extractions and qPCR reactions were carried out following the procedure described by Mayo-Prieto et al. [24]. 250 mg of each soil was used for total DNA extraction. FavorPrep Soil DNA Isolation Kit (Favorgen Biotech Corporation, Ping-Tung, Taiwan) was used following the manufacturer's instructions. A DNA extraction was performed by repetition.

### 2.7. Statistical Analysis

Means and error of the recorded data were calculated to evaluate the development of each *Trichoderma* spp. in different soils. The data were transformed by the formula $(x + 0.5)^{1/2}$ [35].

The fungal growth in AS and NS was analyzed by Levene's test and compared by analysis of variance (three-way ANOVA for a completely randomized design including main effects of *Trichoderma* isolates with two levels, T029 and T059, soil treatment with two levels AS and NS, and localizations from A1 to P5) and post hoc analysis of Tukey's test was done for each *Trichoderma* isolates and soil treatment.

The effects of soil parameters on comparison to *Trichoderma* concentration in both soils were determined by Canonical Correspondence Analyses (CCAs) with the R package CCA software [36]. Dataset of physicochemical components was divided in two dataset (soil parameters and nutrients) and compared to *Trichoderma* data in two CCAs. All data were normalized and standardized prior CCA in order to ensure the multivariate normal distribution. A low standard deviation was achieved and it is not necessary to use standardized coefficients for correlation matrix. Values of raw canonical coefficients were

compared and Wilks' Lambda, using F-approximation was used for statistical analysis of dimensions. Data of correlation were represented in a biplot.

## 3. Results

### 3.1. Standard curve of Trichoderma spp.

The linear regression equation for *T. velutinum* T029 is $y = -2.8717 \, x + 18.120$ and for *T. harzianum* T059 $y = -3.4754 \, x + 18.726$ with a highly significant correlation in both isolates (Figure 1).

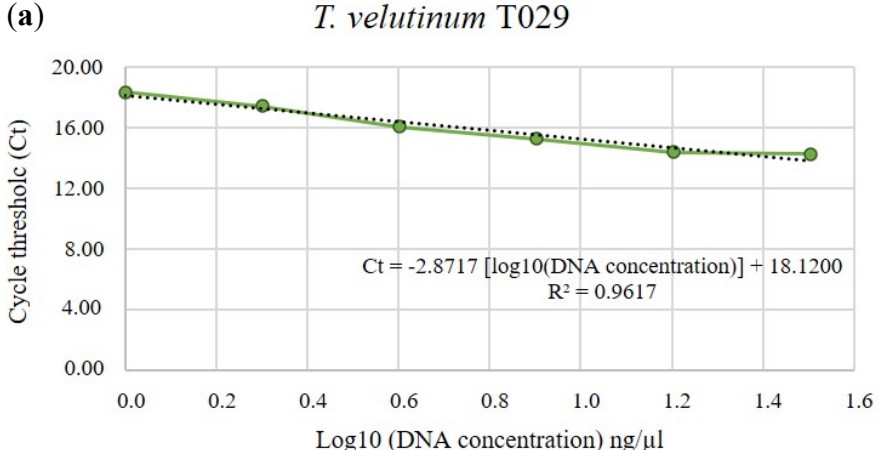

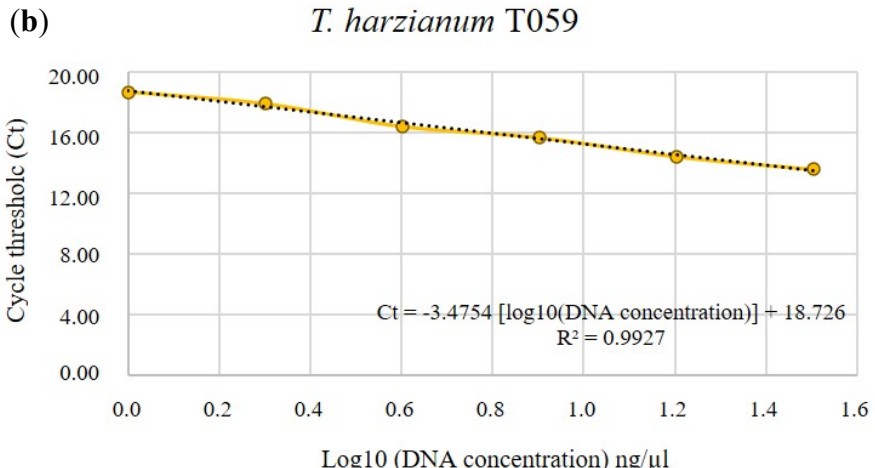

**Figure 1.** Standard curves of *T. velutinum* T029 (**a**) and *T. harzianum* T059 (**b**) DNA concentration standards vs. the cycle threshold (Ct) values from a qPCR. Ct values were plotted against logarithmically transformed DNA amounts ranging from 10 ng to 320 ng of genomic DNA·$\mu L^{-1}$. The linear regression equation is written in every graph with its linear regression coefficient ($R^2$).

### 3.2. Soil Sampling and Physicochemical Analysis

Twelve plots were selected for sampling in the bean production area of León (Table 1). Therefore, one plot was sampled in La Bañeza, two in La Cabrera and Esla-Campos, respectively, three in Astorga, and five in Páramo.

The physicochemical characteristics were different for each soil. Regarding soil texture, in Paramo and La Bañeza, they were sandy loam, and the rest were between loam and silt loam. pH in Astorga was between strongly acidic and slightly acidic, in La Bañeza neutral, in La Cabrera very strongly acidic, in Esla-Campos between strongly acidic and slightly alkaline, and in Paramo between moderately acidic and slightly alkaline (Table 2).

**Table 2.** Physicochemical characteristics of soil samples.

| Id | Sand (%) | Silt (%) | Clay (%) | Texture | pH | CEC (cmol kg$^{-1}$) | EC (dS m$^{-1}$) | OM (%) | C$_{total}$ (%) | N$_{total}$ (%) | C/N | P (ppm) | K (ppm) | Ca (ppm) | Mg (ppm) | K/Mg | Ca/Mg | Na (ppm) | Mn (ppm) | Fe (ppm) | Cu (ppm) | Zn (ppm) | B (ppm) |
|---|---|---|---|---|---|---|---|---|---|---|---|---|---|---|---|---|---|---|---|---|---|---|---|
| A1 | 38 | 54 | 8 | Silt loam | 6.23 | 6.79 | 0.15 | 2.67 | 1.50 | 0.14 | 11.22 | 46.53 | 175.94 | 843.64 | 108.16 | 1.63 | 7.80 | 45.98 | 22.23 | 46.37 | 1.32 | 2.24 | 0.61 |
| A2 | 40 | 52 | 8 | Silt loam | 6.09 | 4.92 | 0.07 | 1.81 | 1.07 | 0.13 | 8.11 | 62.36 | 191.58 | 661.29 | 75.35 | 2.54 | 8.78 | 13.79 | 18.98 | 55.84 | 1.70 | 2.56 | 0.38 |
| A3 | 38 | 50 | 12 | Silt loam | 5.25 | 3.69 | 0.19 | 3.58 | 2.08 | 0.21 | 9.90 | 51.18 | 293.24 | 446.87 | 54.69 | 5.36 | 8.17 | 20.69 | 32.74 | 40.11 | 1.11 | 1.14 | 0.29 |
| B2 | 54 | 38 | 8 | Sandy loam | 6.63 | 4.94 | 0.08 | 1.15 | 0.67 | 0.07 | 8.94 | 28.05 | 78.20 | 733.43 | 86.28 | 0.91 | 8.50 | 11.49 | 19.17 | 35.13 | 0.94 | 0.76 | 0.28 |
| C1 | 40 | 46 | 14 | Loam | 4.58 | 4.01 | 0.30 | 4.31 | 2.47 | 0.21 | 11.81 | 96.67 | 308.88 | 529.03 | 69.27 | 4.46 | 7.64 | 9.20 | 48.53 | 91.69 | 1.30 | 2.18 | 0.25 |
| C2 | 46 | 40 | 14 | Loam | 4.90 | 4.70 | 0.32 | 4.51 | 2.66 | 0.20 | 13.09 | 86.27 | 312.79 | 615.20 | 72.92 | 4.29 | 8.44 | 22.99 | 37.61 | 75.92 | 1.18 | 2.30 | 0.28 |
| E1 | 48 | 32 | 20 | Loam | 7.79 | 8.46 | 0.17 | 1.22 | 0.82 | 0.10 | 7.00 | 31.93 | 152.48 | 2941.73 | 149.48 | 1.02 | 19.68 | 18.39 | 18.63 | 14.20 | 0.63 | 0.66 | 0.43 |
| P1 | 52 | 36 | 12 | Sandy loam | 6.91 | 7.05 | 0.11 | 0.62 | 0.81 | 0.10 | 3.53 | 27.90 | 160.30 | 1128.20 | 132.46 | 1.21 | 8.52 | 13.79 | 32.52 | 33.97 | 0.88 | 0.67 | 0.41 |
| P2 | 58 | 34 | 8 | Sandy loam | 7.45 | 5.52 | 0.05 | 0.27 | 0.45 | 0.07 | 2.25 | 36.90 | 74.29 | 839.63 | 138.54 | 0.54 | 6.06 | 9.20 | 14.74 | 22.11 | 0.64 | 0.96 | 0.50 |
| P3 | 64 | 30 | 6 | Sandy loam | 5.66 | 3.32 | 0.09 | 0.62 | 0.54 | 0.08 | 4.38 | 77.73 | 215.04 | 442.86 | 34.03 | 6.32 | 13.02 | 2.30 | 41.29 | 64.33 | 0.93 | 1.20 | 0.48 |
| P4 | 60 | 30 | 10 | Sandy loam | 6.13 | 7.71 | 0.08 | 0.97 | 0.89 | 0.11 | 4.94 | 52.11 | 187.67 | 1180.30 | 134.89 | 1.39 | 8.75 | 13.79 | 25.73 | 50.15 | 0.55 | 0.92 | 0.50 |
| P5 | 54 | 34 | 12 | Sandy loam | 7.06 | 7.44 | 0.09 | 0.56 | 0.77 | 0.11 | 3.06 | 69.19 | 226.77 | 1102.15 | 115.45 | 1.96 | 9.55 | 13.79 | 23.98 | 31.90 | 0.90 | 1.28 | 0.65 |

Code of the soil sampling used for this study and described in Table 1 (A: Astorga, B: La Bañeza; C: La Cabrera; E: Esla-Campos; P: El Páramo); The results were obtained following the procedure described in Section 2.4. Physicochemical analysis of soils. Texture soil; Cation exchange capacity (CEC), electric conductivity (EC), soil organic matter (OM), nitrogen (N$_{total}$) and carbon (C$_{total}$) content, C:N ratio (C/N), assimilable phosphorus (P), the cations potassium (K), calcium (Ca), magnesium (Mg) and sodium (Na), K:Mg ratio (K/Mg), Ca:Mg ratio (Ca/Mg), and the microelements manganese (Mn), iron (Fe), copper (Cu), zinc (Zn) and boron (B).

In the case of CEC, it was low, between 3.32 cmol·kg$^{-1}$ and 8.46 cmol·kg$^{-1}$ in Paramo (P3) and Esla-Campos (E1), respectively. As for EC, all soils were non-saline with values between 0.05 dS·m$^{-1}$ (P2) and 0.32 dS·m$^{-1}$ (C2). OM concentration in the plot samples was remarkably high in La Cabrera, medium to high in Astorga, medium in La Bañeza, medium to very low in Esla-Campos and low to very low in Páramo, with values between 4.51% (C2) and 0.05% (P2). Regarding to the C/N ratio, it was between medium and high in most of the soils, except for the Paramo soils, in which it was low. P and K concentrations were high in all soils except for the K concentration in Paramo soils, where it was low. Most soils had a low Ca concentration. Mg concentration was high in most soils of Paramo, Astorga, and Esla-Campos, but it was low in the rest of them. All soils presented from low to acceptable concentrations of Na, Mn, Fe, Cu, Zn, and B (Table 2).

### 3.3. Development of Trichoderma spp. in Crop Soils

In the Levene's test, the data did not present significant differences. There were significant differences between *Trichoderma* isolates (T029 and T059), between soil treatment (AS and NS), between localizations (from A1 to P5, Table 1). The interactions of *Trichoderma* isolates x localizations, soil treatments x localizations and, soil treatments x *Trichoderma* isolates x localizations were also significant (Supplementary Material Table S1).

The growth of *Trichoderma* spp. was different depending on the localization and its treatment (AS and NS) (Figure 2). In general, both isolates developed better in AS than in NS. In the AS of La Cabrera (C2), it was detected the highest amount of *T. velutinum* T029 (2.67 µg·g soil$^{-1}$). *T. harzianum* T059 grew the best in the AS of Esla-Campos (E1) (3.13 µg·g soil$^{-1}$). The development of the BCA isolates was more deficient in the NS. In the NS of the Cabrera soil (C2), 1.44 µg·g soil$^{-1}$ of *T. velutinum* T029 was detected.

The development of both isolates was similarly in Astorga and Paramo soils. Both *Trichoderma* isolates had positive growth in both AS and NS soils of this last productive zone. The Cabrera soils (C1, C2) had a negative effect on the *T. harzianum* T059 growth.

### 3.4. Correlations of Trichoderma Isolates and Physicochemical Characteristics of Soil Samples and CCA

In the compared texture of soils, pH, EC and C/N were chosen as elements that could affect to *Trichoderma* growth. Main macronutrients (P, K, Ca and Mg) and micronutrients (Cu and B) were selected as candidates that could induces the survival of these *Trichoderma* species in soils.

The evaluation of these physicochemical parameters of the different bean cultivated soils were divided in two analysis of CCAs for comparing the *Trichoderma* species development. A combination of datasets (soil parameters x *Trichoderma* development) and another combination of datasets (nutrients x *Trichoderma* development). The dataset of *Trichoderma* was the same for both of them.

Wilks' Lambda's test was used for checking the significance of canonical correlation. For the first combination of datasets (soil parameters x *Trichoderma*), the first dimension obtained a *p* value = 0.013 (*p* < 0.05) in soil parameters respect to *Trichoderma*. So that, it indicates a correlation between datasets as well as the second combination of datasets (nutrients x *Trichoderma* development) where a *p* value = 0.011 (*p* < 0.05) indicates a correlation. The rest of dimensions of both combination of datasets showed a p value higher than 0.05. So that, it will be evaluated the correlation for both combination of datasets in the first dimension of the correlation matrix in each combination of datasets.

Combination of the datasets (soil parameters x *Trichoderma* development) are represented in Figure 3a. For the first correlation matrix related to soil parameters, the first dimension was strongly positive related to EC (0.86), clay (0.59) and OM (0.58) and pH (−0.54) a negative correlation was performed. *Trichoderma* data set had a first canonical dimension strongly influenced by AS T029 and NS T029 (0.70 and 0.77 respectively). Figure 4 represents the combination of all values evaluated.

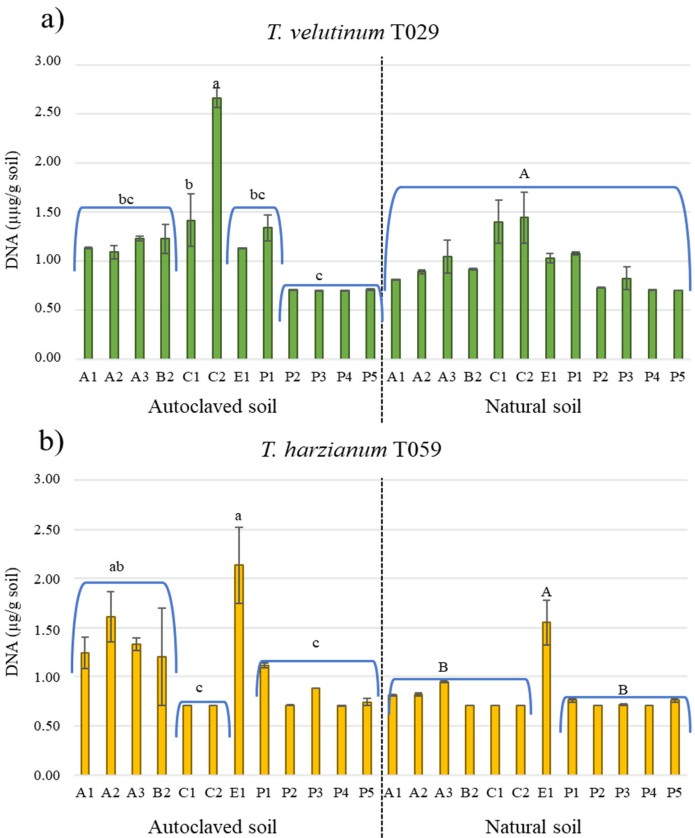

**Figure 2.** DNA concentration (µg·g soil$^{-1}$) of *T. velutinum* T029 (**a**) and *T. harzianum* T059 (**b**) development in different soil samples after 15 days. Left: Autoclaved soil: soil sample autoclaved to 121 °C 20 min. Right: Natural soil, soil sample not autoclaved. Upper and lower error bars are represented and indicated standard error of the mean showing the accuracy of the calculations. Differences statistically significant (Tukey's test $p < 0.05$) are indicated with different letters; capital letter are differences in natural soils (NS) and small letter are differences between autoclaved soils (AS). The code of each soil is detailed in Table 1.

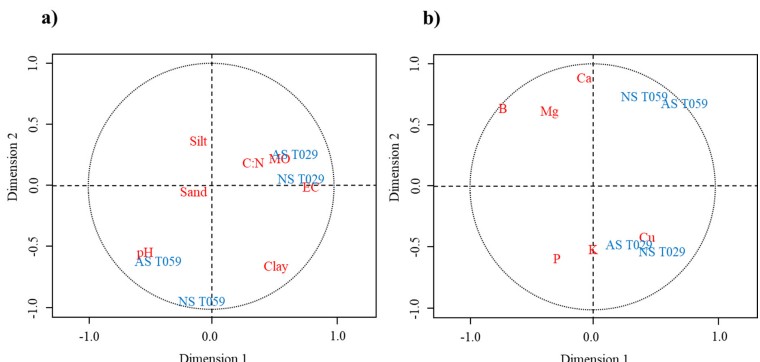

**Figure 3.** Canonical correlation analysis (CCA) based on soil parameters. (**a**) Diagram of soil parameters: sand, silt, clay, pH, electric conductivity (EC), soil organic matter (OM), C:N ratio (C:N). (**b**) Diagram of soil elements: assimilable phosphorus (P), cation potassium (K), calcium (Ca), magnesium (Mg), copper (Cu) and boron (B). Both diagrams *Trichoderma* development: natural soil *T. harzianum* (NS T059), natural soil *T. velutinum* (NS T029), autoclaved soil *T. harzianum* (AS T059) and autoclaved soil *T. velutinum* (AS T029).

The other combination of datasets (nutrients x *Trichoderma* development) are visualized in Figure 3b. According to nutrients, the first canonical dimension was most strongly influenced by AS T059 (0.73). The nutrients in the first dimension was B (−0.64), and Cu (0.55). Figure 5 represents the degree of correlation for all the values represented for the combination *Trichoderma* x nutrients.

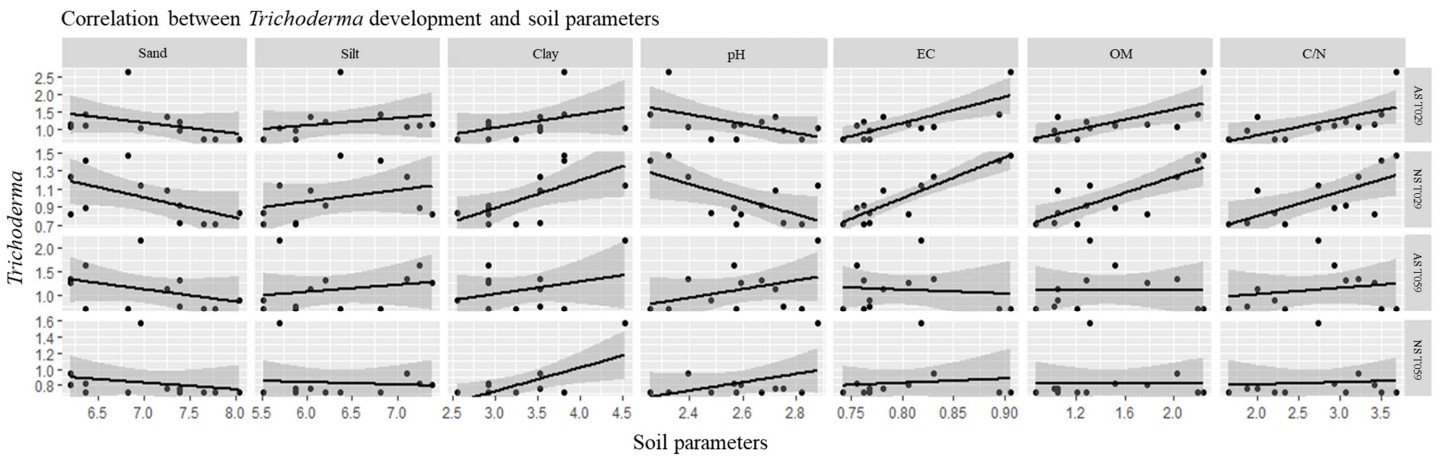

**Figure 4.** Correlation values of the Canonical Correlation Analysis (CCA) between *Trichoderma* spp. and soil parameters. Natural soil *T. harzianum* (NS T059), natural soil *T. velutinum* (NS T029), autoclaved soil *T. harzianum* (AS T059). autoclaved soil *T. velutinum* (AS T029). Electric conductivity (EC), Soil organic matter (OM), C:N ratio (C/N).

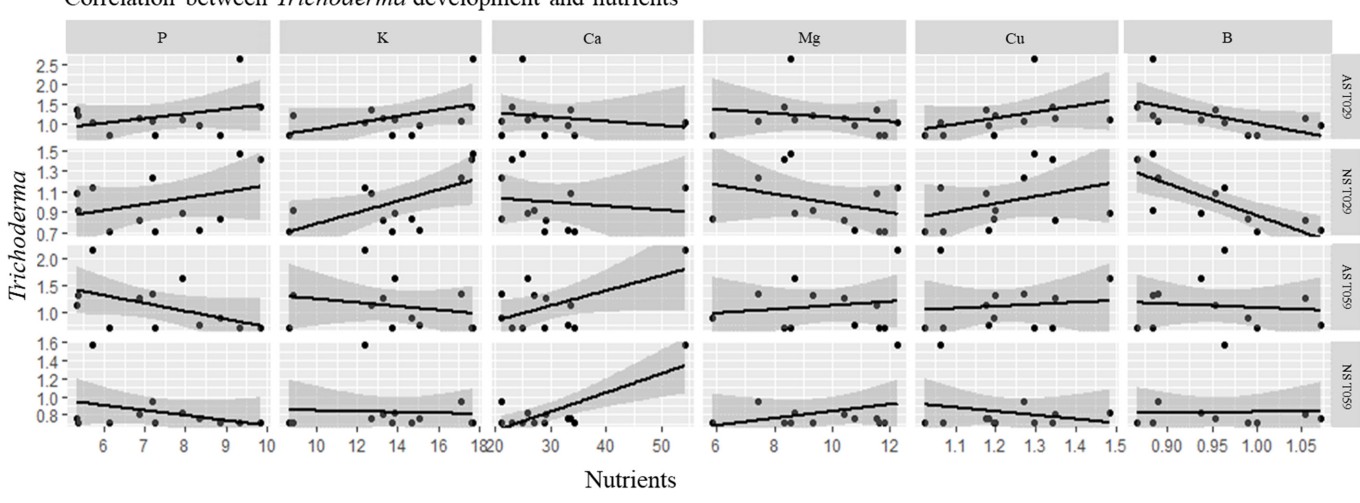

**Figure 5.** Correlation values of the Canonical Correlation Analysis (CCA) between *Trichoderma* spp. and nutrients. Natural soil *T. harzianum* (NS T059), natural soil *T. velutinum* (NS T029), autoclaved soil *T. harzianum* (AS T059). autoclaved soil *T. velutinum* (AS T029). Assimilable phosphorus (P), the cations potassium (K), calcium (Ca), magnesium (Mg), copper (Cu) and boron (B).

The three datasets represented in these analyses (soil parameters, nutrients and *Trichoderma* development) were also evaluated separately and a correlation between their own elements was also found. They are described in Supplementary Material Figures S1–S3. *Trichoderma* dataset, there was a high correlation between AS and NS in each *Trichoderma*, *T. velutinum* T029 (0.756) and *T. harzianum* T059 (0.828) (Supplementary Figure S1). Soil parameter dataset, OM had a very high correlation with EC (0.876) and C:N ratio (0.923), silt with sand (−0.918) (Supplementary Figure S2). Nutrient dataset, P and K had a high correlation (0.797) and Ca and Mg (0.773) (Supplementary Figure S3).

## 4. Discussion

The number of chemically-synthesized products for the control of phytopathogens has been reduced by the increase in food and environmental safety. Agricultural production is nowadays oriented toward sustainable processes that favor local development, using of local varieties, and being environmentally friendly. One way to carry out this process is by using biological agents to control pests and diseases. We have selected two *Trichoderma* isolates, *T. velutinum* T029 and *T. harzianum* T059, obtained from the PGI area where farmers grow adapted and traditional varieties, such as "Canela", "Riñón", "Pinta" and "Plancheta". The use of autochthonous BCAs can be more accurate as they are better adapted than other isolates from other areas and crops. Thus, these isolates are adapted to the agronomic conditions of the field, and they should be more effective in protecting bean plants against biotic factors such as fungal diseases.

In this work, we used a qPCR strategy for the quantification of *Trichoderma* spp. development. It allowed us to know the amount of the biocontrol agent grown in each soil by an objective and accurate method to determine fungal growth. This technique has been previously pursued to quantify the bacteria and soil-borne fungal pathogens in strawberry [37] as well as to detect *Rhizoctonia solani* in tobacco fields, where they quantified the potential inoculum of this pathogen and its disease index [38]. *Trichoderma* was quantified following this method to determine the best horticultural substrate for its development, favouring phytopathogen displacement [24].

*Trichoderma*, as a BCA, can efficiently colonize the rhizosphere, helping plant protection in the presence of pathogens. This process is critical for the interaction with plants and the suppression of soil-borne diseases. It depends on a plethora of biotic and abiotic factors. In this research, *T. velutinum* T029 and *T. harzianum* T059 developed significantly better in autoclaved soils, compared to natural soils. These differences in growth showed between soils were similar to those observed in other microorganisms. Actually, *Bacillus subtilis* presented a higher population in an autoclaved substrate with cottonseeds than in a non-autoclaved one, especially 14 days after inoculation [39]. In another work, peanut plants presented more colonization of *Glomus* spp. in autoclaved soil enriched with soil microbiota than in a non-autoclaved soil [40]. According to several authors, a cause for this favored development could be attributable to the diminished competitive activity of soil microorganisms. During the autoclaving process, competing microorganisms have been removed, some of which may have been autochthonous fungal isolates that were not as effective as the introduced fungus [41]. These indigenous species compete for the nutrients and niche with *Trichoderma* or inhibit its growth by the production of antimicrobial metabolites [39]. Many fungi, bacteria, insects, nematodes, and other microorganism coexist in the soil, so it is particularly important to understand those interactions to develop a soil management strategy, instead of focusing on individual disease-causing species. In a NS, *Trichoderma* spp. might be attacked by mycoparasites that can play a role in limiting fungal populations and have possible effects on plant growth [42]. Al-Khaliel [40] suggested that the autoclaving process increases nutrient availability.

Rhizosphere microorganisms influence the dynamics of the soil organic matter and nutrient cycles [43,44]. Likewise, the development of microorganisms, like *Trichoderma*, is swayed by the characteristics of the soil (Figure 2). Latour et al. [45] investigated the diversity of the populations of *Pseudomonas fluorescens* associated with tomato and flax in

two different soils. They observed that the substrate was the main factor responsible for the heterogenicity of the associated bacterial populations. In our research, the development of *Trichoderma* varied according to the plot. In Paramo's soils, none of our *Trichoderma* isolates developed extensively. Previous reports indicate that pH, C concentration, and EC have a strong influence on soil microbial community composition and function [46–48]. In the present work, Paramo's soils had low total C concentration (between 0.40% and 0.89%), and the EC was the lowest value of the sampled soils, affecting the development of *Trichoderma* spp. The pH value also affected the BCA growth, since it showed a different effect in each productive zone and each sample soil. These data are supported by the research of Brockett et al. [49], who observed similar results. However, in another study, pH affected the quantity of inoculum of *R. solani*, being the range of pH of 4.5–6.5 optimal for its development. Further, if pH increased, the amount of inoculum diminished [38]. *T. velutinum* T029 developed more in La Cabrera's soils than in other soil samples. These soils were characterized for having high levels of OM, total C, C:N ratio, P, K, Fe, and Zn. However, *T. harzianum* T059 grew better on the Esla-Campos's soils, where clay, CEC, pH, Ca, Mg, and Mn showed the highest values of the sampled soils. Harries et al. [38] observed that the development of *R. solani* was limited in soils with high clay, OM content, and physical stability. Also, the abundance of *Blastocladiosmycota phylum, Nitrospirae* or *Acidobacteria phylum*, was affected by pH, total C, total or available P, and N, being all these parameters negative for their development [50]. In our study, *T. harzianum* T059 grew better in soils with a high concentration of clays. An explanation could be that small particles size provide small pore sizes and protect microorganisms against other organisms. In addition, silt and clay particles have more water holding capability and exert a greater impact on water and nutrient availability [50]. *T. velutinum* T029 developed better in soils with high levels of OM, total C and C:N ratio. The addition of organic matter or mineral fertilizers affects microbial activities. Gorissen et al. [51] found that the application of nitrogen to the soil negatively affected the bacterial population associated with the roots of *Pseudotsuga menziesii*. Similarly, Demoling et al. [52] observed that the total C and N levels influenced microbial communities. A high C:N ratio is characteristic of soil systems with a high amount of organic matter and a slow decomposition rate, which should have a negative impact in bacterial and fungal development [11]. Summarizing, C:N ratio and pH, were the main factors that explain the variation in microorganism communities. Furthermore, these characteristics in combination with texture and EC could have influenced the development of *Trichoderma*, regardless of the influence of vegetation type and land-use practices.

The biocontrol activity of the *Trichoderma koningii* strain was also influenced by physical and chemical parameters in the soil, such as iron, copper, texture and boron in a positive correlation and pH and available phosphorus negatively [53]. Some of the nutrients and physical parameters are correlated to some of our strains such as pH with a negative influence and copper and boron positive correlation. This demonstrates the importance of the characteristics of soils that share some of these parameters but that are different from the *Trichoderma* species, using the method of evaluation and crops that are described [54].

All these variations influence the success of a *Trichoderma* strain for being established in a novel soil. So, a physical and chemical analysis is compulsory before applying a *Trichoderma* strain, in order to ensure its positive effects.

## 5. Conclusions

Twelve plots were selected for sampling in the production area of bean crops in the León province. The physicochemical characteristics indicated that the soil texture was between sandy loam and silt loam. The pH was between strongly acidic and slightly alkaline. The OM concentration was between low and remarkably high. The total C and N concentrations and their ratio were between medium and high in most of the soils. The rest of the micronutrients had an acceptable concentration except for Paramo's plots. *T. velutinum* T029 and *T. harzianum* T059 developed better in AS than in NS. High levels

of OM, total C, C:N ratio, P, K, Fe, and Zn in La Cabrera's soils favored the growth of *T. velutinum* T029. The high values of clay, CEC, pH, Ca, Mg, and Mn concentration in Esla-Campos's soils favored the development of *T. harzianum* T059. Texture, particularly clay concentration, OM, EC, and pH in the soil parameters and B and Cu in the nutrients could be the main factors explaining the influence in the *Trichoderma* development. The OM, EC, and C:N ratio and Cu are the main soil characteristics that influence in *T. velutinum* T029 development and pH in the *T. harzianum* T059 development.

**Supplementary Materials:** The following are available online at https://www.mdpi.com/2073-4 395/11/2/274/s1, Table S1: Three-way ANOVA for Trichoderma DNA concentration ($\mu g \cdot g$ soil$^{-1}$), Figure S1: Correlation values of the Canonical Correlation Analysis (CCA) between isolates development, *Trichoderma velutinum* T029 and *T. harzianum* T059, in the soil treatment. Natural soil *T. harzianum* (NS T059), natural soil *T. velutinum* (NS T029), autoclaved soil *T. harzianum* (AS T059). autoclaved soil *T. velutinum* (AS T029), Figure S2: Correlation values of the Canonical Correlation Analysis (CCA) between soil parameters. Electric conductivity (EC), Soil organic matter (OM), C:N ratio (C/N), Figure S3: Correlation values of the Canonical Correlation Analysis (CCA) between nutrients. Assimilable phosphorus (P), the cations potassium (K), calcium (Ca), magnesium (Mg), copper (Cu) and boron (B).

**Author Contributions:** All authors contributed to the study conception and design. Material preparation, sampling and data collection were performed by S.M.-P., A.J.P.-Á., S.d.S.-H., Á.R.-G., and G.C.-H.; statistical analysis was carried out by S.M.-G., G.C.-H. and S.M.-P.; supervision of all study was performed by S.G. and P.A.C.; the first draft of the manuscript was written by S.M.-P. and all authors commented on previous versions of the manuscript. All authors read and approved the final manuscript.

**Funding:** This research was funded by Junta de Castilla y León, Consejería de Educación for the project "Application of *Trichoderma* strains in sustainable quality bean production" (LE251P18).

**Conflicts of Interest:** The authors declare no conflict of interest. The funders had no role in the design of the study; in the collection, analyses, or interpretation of data; in the writing of the manuscript, or in the decision to publish the results.

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
