# Peer review of "Influence of Physicochemical Characteristics of Bean Crop Soil in Trichoderma spp. Development"

_agronomy, doi:10.3390/agronomy11020274_

Round 1

Reviewer 1 Report

The article is potentially written deciphering the “Influence of physicochemical characteristics of bean crop soil in Trichoderma spp. development” by Mayo-Prieto et al. However, the manuscript has to be improved to make it more suitable for acceptance in the journal. The following questions need to be addressed.

Comment 1: In the abstract, did not find single information about bean/bean crop soil?

Comment 2: line no. 153, rewrite the sentence.

Comment 3: Give footnote for Tables.

Comment 4: Please change unit ug/g soil to μg/g using the insert symbol option. Please change figure captions as Fig. 1a…..and Fig. 1b…instead up the green color and down yellow col.

Comment 5: Please change Table 3 format. Please check the author instructions.

Comment 6: Figure 2, captions not clear. Please rewrite.

Comment 7: line no. 304 sandy loam and…

Comment 7: Please check the typographical error and English throughout the manuscript.

Author Response

The article is potentially written deciphering the “Influence of physicochemical characteristics of bean crop soil in Trichoderma spp. development” by Mayo-Prieto et al. However, the manuscript has to be improved to make it more suitable for acceptance in the journal. The following questions need to be addressed.

Comment 1: In the abstract, did not find single information about bean/bean crop soil?

We added the sentence (Lines 18-20) “The soils of this area have between silt loam and sandy loam texture, with moderate clay content, neutral or acidic pH, rich in organic matter and low carbonate levels, providing beans with high water absorption capacity and better organoleptic qualities after cooking”.

Comment 2: line no. 153, rewrite the sentence.

We rewritten the sentence: “A DNA extraction was performed by repetition.”

Comment 3: Give footnote for Tables.

We added in the Table 2:

“1. Code of the soil sampling used for this study and described in Table 1 (A: Astorga, B: La Bañeza; C: La Cabrera; E: Esla-Campos; P: El Páramo). The results were obtained following the procedure described in section 2.4. Physicochemical analysis of soils.

  1. Texture soil

Cation exchange capacity (CEC), electric conductivity (EC), soil organic matter (OM), nitrogen (total N) and carbon (Total C) content, C:N ratio (C:N), assimilable phosphorus (P), the cations potassium (K), calcium (Ca), magnesium (Mg) and sodium (Na), K:Mg ratio (K:Mg), Ca:Mg ratio (Ca:Mg), and the microelements manganese (Mn), iron (Fe), copper (Cu), zinc (Zn) and boron (B)”.

And in the Table 3:

  1. Cation Exchange Capacity (CEC); electric conductivity (EC); 2 Soil organic matter (OM), nitrogen (total N) and carbon (Total C) content, C:N ratio (C/N), assimilable phosphorus (P), the cations potassium (K), calcium (Ca), magnesium (Mg) and sodium (Na), K:Mg ratio (K/Mg), Ca:Mg ratio (Ca/Mg), and the microelements manganese (Mn), iron (Fe), copper (Cu), zinc (Zn) and boron (B)”

Comment 4: Please change unit ug/g soil to μg/g using the insert symbol option. Please change figure captions as Fig. 1a…..and Fig. 1b…instead up the green color and down yellow col.

We corrected these mistakes and rewritten the figure captions.

Comment 5: Please change Table 3 format. Please check the author instructions.

We changed the commas for the periods to adapt it to the standards of the journal.

Comment 6: Figure 2, captions not clear. Please rewrite.

We corrected these mistakes and rewritten the figure captions

Comment 7: line no. 304 sandy loam and…

We corrected this mistake. Thanks for the correction.

Comment 7: Please check the typographical error and English throughout the manuscript.

The English writing has been completely revised.

Reviewer 2 Report

The authors investigated the relative importance of various physiochemical parameters of the soil collected from their study sites, in the adaptation of Trichoderma spp. The research targets determining the best horticultural substrate for the development of biocontrol agents such as Trichoderma, and in turn manage phytopathogen to control diseases on crops like beans. The researchers have conceived and executed the work, but the reviewer has some concerns about the English language and style. The authors also present some aspects relevant to methodology in the results section and make contradictory statements in the results and discussion.  

Few minor and major comment are provided below:

Minor comments below relate to correcting the language and style:

Line 108: For the qPCR reactions, it was used the Step One Plus™
Line 121: it was used the wet determination
Line 141: to eliminate any present organism
Lines 158-162: lacks clarity; suggest rephrase.

Lines 166-168, 173-175: move these lines to methodology

Line 175: the surface dedicated to the bean production?surface area?

Major comments:

Table 2 and Table 3 uses different formatting for decimal values (comma versus period). Follow the guidelines of the Agronomy journal.

Line 276 states the pH value did not affect the BCA growth and
provide supporting arguments, but then line 299 makes
contradictory statements.

Author Response

The authors investigated the relative importance of various physiochemical parameters of the soil collected from their study sites, in the adaptation of Trichoderma spp. The research targets determining the best horticultural substrate for the development of biocontrol agents such as Trichoderma, and in turn manage phytopathogen to control diseases on crops like beans. The researchers have conceived and executed the work, but the reviewer has some concerns about the English language and style. The authors also present some aspects relevant to methodology in the results section and make contradictory statements in the results and discussion. 

Few minor and major comment are provided below:

Minor comments below relate to correcting the language and style:

Line 108: For the qPCR reactions, it was used the Step One Plus™

We corrected “Step One Plus™ (Applied Biosystems) device and α-actin as the reference gene for all analyses were used for the qPCR reactions, according to the procedure described by Mayo-Prieto et al”.

Line 121: it was used the wet determination

We rewritten “For the determination of the textural class, it was used the Bouyoucos densimeter method and graphically represented by the triangle of the United States Department of Agriculture (USDA”

Line 141: to eliminate any present organism

We corrected it “The AS was autoclaved at 121 ºC 20 min eliminating any present microorganism.”

Lines 158-162: lacks clarity; suggest rephrase.

This part described the steps in the R software and it usually writes in this way. So, we did not change these sentences.

Lines 166-168, 173-175: move these lines to methodology

We rewritten these sentences “Lines 117-118 (Material and methods): “The number of samples collected in each production area was proportional to the area dedicated to the bean production.”

We cancelled the sentence of the lines 173-175, remaining Lines 170-174: “The linear regression equation for T. velutinum T029 is  and for T. harzianum T059  with a highly significant correlation in both isolates (Figure 1).”

Line 175: the surface dedicated to the bean production? surface area?

We referred to the area dedicated to the bean production.

Major comments:

Table 2 and Table 3 uses different formatting for decimal values (comma versus period). Follow the guidelines of the Agronomy journal.

We changed the commas for the periods to adapt it to the standards of the journal.

Line 276 states the pH value did not affect the BCA growth and provide supporting arguments, but then line 299 makes contradictory statements.

We corrected this mistake. Line 282 “The pH value also affected the BCA growth, since it showed a different effect in each productive zone and each sample soil.”

Reviewer 3 Report

The topic is very interesting and actual: use Trichoderma spp. as biological agent to reduce fungicide applications in agriculture.
The experimental design is appropriate (soil sampling and 3 replications) as well as the use of Real-Time PCR.
However, the data analyses ANOVA and PCA deserve major comments.
ANOVA:
Authors did not mention the ANOVA model that was applied: ANOVA1 or ANOVA2 including 2 main factors, Thrichoderma species (2 levels: T029 and T059) and Soil (2 levels: AS and NS), and interaction Species x Soil.
Many treatments have zero or very low mean values. These should have been excluded from the ANOVA because of the lack of homogeneity of variances among treatments, applying an ANOVA1 comparing only those treatments (Species-Soil combinations) where the growth of Trichoderma spp. was acceptable. Moreover, based on the bars shown in the Figure (named Figure1 but it was Figure2 in the text) a strong variation among samples was found. The Authors did not use the Bartlett's test to check for homogeneity of error variances.
The Fisher LSD test for multiple comparisons is fine but the Tukey's HSD test would have been preferable. However, in Figure 1 (Figure 2) it is not clear why the Authors carried out 2 separate groups of tests among means, as reported in the Figure's legend "Differences statistically significant (p< 0.05) are indicated with different letters; capital letter are differences between all soils and small letter are differences between every soil (autoclaved soil and natural soil)". Multiple comparisons among treatment means include all of them in a single analysis.

In Figure 1 (Figure2 in the text) the combinations of E2 (site Jabares de los Oteros) with both Trichoderma species are NOT shown. This lack is not mentioned in the text. Why it does not appear in the Figure? If there is a reason the Authors must explain it. Moreover, in Figure caption the DNA concentration is expressed as "ng g soil-1", whereas in the rows 200-205 it is expressed as "µg g soil-1".

"PCA":
Authors applied Principal Component Analysis "to observe how the variations on the soil 158 characteristics affect the development of both Trichoderma". However, Principal Component Analysis can be applied only to identify new multivariate coordinates that maximizes the differences among samples. For the purpose of this research, Authors must have used a Canonical Correlation approach. In fact, the figure showing the "PCA" results (Figure2 but it was Figure 3 in the text) is quite confusing. In particular, it is not clear how the Authors related the soil variables with the 4 combinations of Trichoderma spp. and Soil, and how these 4 categories could be grouped in just one quadrant. These results, if correct, must be explained because these 4 categories (Trichoderma spp. x Soil) seems very similar on average among each other but the response of Trichoderma growth was very different.

A more in depth explanation of the statistical approach applied and of the discussion of the results shown in rows 220-227 must be carried out to understand how and which soil parameters influenced the growth of Tricoderma in both autoclaved and natural soils.

Author Response

The topic is very interesting and actual: use Trichoderma spp. as biological agent to reduce fungicide applications in agriculture.

The experimental design is appropriate (soil sampling and 3 replications) as well as the use of Real-Time PCR.

However, the data analyses ANOVA and PCA deserve major comments.

ANOVA:

Authors did not mention the ANOVA model that was applied: ANOVA1 or ANOVA2 including 2 main factors, Trichoderma species (2 levels: T029 and T059) and Soil (2 levels: AS and NS), and interaction Species x Soil.

Many treatments have zero or very low mean values. These should have been excluded from the ANOVA because of the lack of homogeneity of variances among treatments, applying an ANOVA1 comparing only those treatments (Species-Soil combinations) where the growth of Trichoderma spp. was acceptable. Moreover, based on the bars shown in the Figure (named Figure1 but it was Figure2 in the text) a strong variation among samples was found. The Authors did not use the Bartlett's test to check for homogeneity of error variances.

We have rewritten the section of statistical analysis: “Means and error of the recorded data were calculated to evaluate the development of each Trichoderma spp. in different soils. The data were transformed by the formula (x + 0.5)1/2 to fit a normal distribution [35]. The fungal growth in AS and NS was analyzed by Levene’s test and compared by analysis of variance (One-way ANOVA) and Tukey’s test”.

The Fisher LSD test for multiple comparisons is fine but the Tukey's HSD test would have been preferable. However, in Figure 1 (Figure 2) it is not clear why the Authors carried out 2 separate groups of tests among means, as reported in the Figure's legend "Differences statistically significant (p< 0.05) are indicated with different letters; capital letter are differences between all soils and small letter are differences between every soil (autoclaved soil and natural soil)". Multiple comparisons among treatment means include all of them in a single analysis.

In Figure 1 (Figure2 in the text) the combinations of E2 (site Jabares de los Oteros) with both Trichoderma species are NOT shown. This lack is not mentioned in the text. Why it does not appear in the Figure? If there is a reason the Authors must explain it. Moreover, in Figure caption the DNA concentration is expressed as "ng g soil-1", whereas in the rows 200-205 it is expressed as "µg g soil-1".

We remade the figure 2 with results of Tukey’s test.

We corrected the Figure’s legend. “Figure 2: : DNA concentration (µg·g soil-1) of T. velutinum T029 (Fig. 2a) and T. harzianum T059 (Fig. 2b) development in different soil samples after 15 days. Left: Autoclaved soil: soil sample autoclaved to 121 ºC 20 min. Right: Natural soil, soil sample not autoclaved. Upper and lower error bars are represented and indicated standard error of the mean showing the accuracy of the calculations. Differences statistically significant (Tukey’s test p< 0.05) are indicated with different letters; capital letter are differences between all soils and small letter are differences between every soil (on the one hand autoclaved soil and on the other natural soil). The code of each soil is detailed in Table 1. E2 soil was only tested as natural soil.”

We added in the text and the Figure 2: “E2 soil was only tested as NS, as there was not enough quantity to evaluate it as AS”. We corrected this mistake: “DNA concentration (µg·g soil-1) of T. velutinum T029 (Fig. 2a) and T. harzianum T059 (Fig. 2b)”

"PCA":

Authors applied Principal Component Analysis "to observe how the variations on the soil 158 characteristics affect the development of both Trichoderma". However, Principal Component Analysis can be applied only to identify new multivariate coordinates that maximizes the differences among samples. For the purpose of this research, Authors must have used a Canonical Correlation approach. In fact, the figure showing the "PCA" results (Figure2 but it was Figure 3 in the text) is quite confusing. In particular, it is not clear how the Authors related the soil variables with the 4 combinations of Trichoderma spp. and Soil, and how these 4 categories could be grouped in just one quadrant. These results, if correct, must be explained because these 4 categories (Trichoderma spp. x Soil) seem very similar on average among each other but the response of Trichoderma growth was very different.

response of Trichoderma growth was very different.

We have applied PCA to identify new multivariate coordinates that maximizes and resumes the differences among samples. The two principal components resume the variation (62.59 %) of soil parameters.  This variation is represented in Figure 3. Figure 3 a) shows Diagram of vectors projection of soil parameters. Figure 3 b) shows Diagram ordination of Trichoderma isolates. We have rewritten the figure’s legend.

Figure 3: Principal component analysis (PCA) based on soils parameters. a) Diagram of vectors projection of soil parameters: sand, silt, clay, pH, cation exchange capacity (CEC), electric conductivity (EC), soil organic matter (OM), nitrogen (total N) and carbon (Total C) content, C:N ratio (C:N), assimilable phosphorus (P), the cations potassium (K), calcium (Ca), magnesium (Mg) and sodium (Na), K:Mg ratio (K:Mg), Ca:Mg ratio (Ca:Mg), and the microelements manganese (Mn), iron (Fe), copper (Cu), zinc (Zn) and boron (B). b) Diagram ordination: natural soil T. harzianum (NS T059), natural soil T. velutinum (NS T029), autoclaved soil T. harzianum (AS T059). autoclaved soil T. velutinum (AS T029).

A more in depth explanation of the statistical approach applied and of the discussion of the results shown in rows 220-227 must be carried out to understand how and which soil parameters influenced the growth of Trichoderma in both autoclaved and natural soils.

We have improved the explanation of the statistical approach applied in rows 220-227.

Round 2

Reviewer 3 Report

ANOVA

The authors explained why they did not include E2 soil as AS because of the lack of sufficient amount of soil. However, in this case, Authors have “missing treatments”, not missing values, in the ANOVA.
I suggest to EXCLUDE E2 from the analysis and rerun an ANOVA2 for a completely randomized design, based on the model including main effects of Trichoderma species (2 levels: T029 and T059) and Soil treatment (2 levels: Autoclaved and Natural) together with the interaction between T. species and Soil treatment. The statement in the legend of Figure 2 “…….; capital letter are differences between all soils and small letter are differences between every soil (on the one hand autoclaved soil and on the other natural soil” is not clear about which comparisons were done. Authors applied Levene’s test to transformed data but they must discuss if it was significant or not.

PCA

Authors did not mention the standardization of data before performing PCA, that is necessary if the different variables show very different variances. Moreover, Authors refer to the use of R by  “estim_ncpPCA function ("Kfold method" for 166 cross-validation); imputation of the data set with the impute”. However, this procedure refers to the estimation of missing values. Authors must mention in the Materials and Methods and in the Results if missing values were in the original data file and at which extent.

Authors again show results of a Principal Component Analysis (PCA) to test for effects of soil parameters on the growth of Trichoderma species under 2 different soil treatments (AS and NS).
However, in this experiment Authors have TWO different groups of variables, one related to soil parameters and one group related to Trichoderma growth.

PCA can be applied when ONE SET of variables is under study.

Authors could apply PCA to evaluate soil differences among the 13 locations where soil samples were collected (for example using the data shown in Table 2).

Authors could also apply PCA to evaluate the growth of Trichoderma on soils sampled in the 13 locations (here again E2 must be removed because of missing treatments AST029 and AST059).

If Authors want to relate soil variables to Trichoderma growth, they must use other multivariate approaches such as Canonical Correlation Analysis.

However, if Authors decide to consider PCA as a correct approach, they must show a Table with the eigenvector coefficients of the variables they considered for the PCA, maybe as a Supplementary Table.

At the end of the abstract, Authors suggest relationships between soil variables and Trichoderma species performance. These conclusions are not discussed in the results referring to Figure3. Many more soil parameters seem to show high correlation values (based on Figure 3a) with both PC1 and PC2 than those mentioned in the abstract. The relative effect of soil parameters on Trichoderma is discussed mainly on the data shown in Table 2  and on results shown by other references, rather than on PCA results.

Finally, having all four Trichoderma-Soil Treatment classification categories in one quadrant is unusual for a PCA.

I suggest Authors to re-evaluate the multivariate approach applied to find a relationship between soil parameters and Trichoderma growth.

Author Response

ANOVA

The authors explained why they did not include E2 soil as AS because of the lack of sufficient amount of soil. However, in this case, Authors have “missing treatments”, not missing values, in the ANOVA.

I suggest to EXCLUDE E2 from the analysis and rerun an ANOVA2 for a completely randomized design, based on the model including main effects of Trichoderma species (2 levels: T029 and T059) and Soil treatment (2 levels: Autoclaved and Natural) together with the interaction between T. species and Soil treatment.

We have excluded E2 and we have repeated all analysis.

We have rewritten the section of statistics analysis “Means and error of the recorded data were calculated to evaluate the development of each Trichoderma spp. in different soils. The data were transformed by the formula (x + 0.5)1/2 to fit a normal distribution [35]. The fungal growth in AS and NS was analyzed by Levene’s test and compared by analysis of variance (two-way ANOVA for a completely randomized design including main effects of Trichoderma isolates with two levels, T029 and T059, and soil treatment with two levels AS and NS) and post hoc analysis of Tukey’s test was done for each Trichoderma isolates and soil treatment.

There was not significant differences between Trichoderma isolates (T029 and T059).. However, there was significantly differences between soil treatments (AS and NS). The interaction Trichoderma isolate x soil treatment was not significant.

The statement in the legend of Figure 2 “…….; capital letter are differences between all soils and small letter are differences between every soil (on the one hand autoclaved soil and on the other natural soil” is not clear about which comparisons were done.

One-way ANOVA, for a completely randomized design including main effects of soil samples and post hoc analysis of Tukey’s test, was done for each Trichoderma isolate and soil treatment.

Figure 2: DNA concentration (µg·g soil-1) of T. velutinum T029 (Fig. 2a) and T. harzianum T059 (Fig. 2b) development in different soil samples after 15 days. Left: Autoclaved soil: soil sample autoclaved to 121 ºC 20 min. Right: Natural soil, soil sample not autoclaved. Upper and lower error bars are represented and indicated standard error of the mean showing the accuracy of the calculations. Differences statistically significant (Tukey’s test p< 0.05) are indicated with different letters; capital letter are differences in natural soils (NS) and small letter are differences between autoclaved soils (AS). The code of each soil is detailed in Table 1.

Authors applied Levene’s test to transformed data, but they must discuss if it was significant or not.

We have added “In the Levene´s test, the data did not present significant differences.

There was not significant differences between Trichoderma isolates (T029 and T059). However, there was significantly differences between soil treatments s (AS and NS). The interaction Trichoderma isolate x soil treatment was not significant.

PCA

Authors did not mention the standardization of data before performing PCA, that is necessary if the different variables show very different variances. Moreover, Authors refer to the use of R by “estim_ncpPCA function ("Kfold method" for 166 cross-validation); imputation of the data set with the impute”. However, this procedure refers to the estimation of missing values. Authors must mention in the Materials and Methods and in the Results if missing values were in the original data file and at which extent.

Authors again show results of a Principal Component Analysis (PCA) to test for effects of soil parameters on the growth of Trichoderma species under 2 different soil treatments (AS and NS). However, in this experiment Authors have TWO different groups of variables, one related to soil parameters and one group related to Trichoderma growth.

PCA can be applied when ONE SET of variables is under study.

Authors could apply PCA to evaluate soil differences among the 13 locations where soil samples were collected (for example using the data shown in Table 2).

Authors could also apply PCA to evaluate the growth of Trichoderma on soils sampled in the 13 locations (here again E2 must be removed because of missing treatments AST029 and AST059).

If Authors want to relate soil variables to Trichoderma growth, they must use other multivariate approaches such as Canonical Correlation Analysis.

However, if Authors decide to consider PCA as a correct approach, they must show a Table with the eigenvector coefficients of the variables they considered for the PCA, maybe as a Supplementary Table.

At the end of the abstract, Authors suggest relationships between soil variables and Trichoderma species performance. These conclusions are not discussed in the results referring to Figure3. Many more soil parameters seem to show high correlation values (based on Figure 3a) with both PC1 and PC2 than those mentioned in the abstract. The relative effect of soil parameters on Trichoderma is discussed mainly on the data shown in Table 2 and on results shown by other references, rather than on PCA results.

Finally, having all four Trichoderma-Soil Treatment classification categories in one quadrant is unusual for a PCA.

I suggest Authors to re-evaluate the multivariate approach applied to find a relationship between soil parameters and Trichoderma growth.

Following the reviewer's suggestions, we have re-evaluated the analysis. We have remade the Figure 3 with the news results and added Fig.3c with a diagram ordination of soil samples.

Round 3

Reviewer 3 Report

ANOVA

Authors excluded the E2 location and carried out an ANOVA2.

The Table 1 and the description of sites where soil samples were collected must be in the Materials and Methods section, not in the results.

However, as shown in Figure 2, the ANOVA has 3 main factors: Thricoderma species (T029 and T059), Soil Treatment (Autoclaved vs Natural) and Locations (from A1 to P5). Therefore, it is a 3 way ANOVA and Figure 2 shows the second order interaction (Species x Soil Treatment x Location). Therefore, in Materials and Methods the Authors should say 3way-ANOVA. Moreover, an ANOVA Table must be added as supplementary table to evidence the significance of variances for each source of variation: main effects and interactions.

PCA

Authors did not mention the standardization of data before performing PCA, that is necessary if the different variables show very different variances. Moreover, Authors refer to the use of R by  “estim_ncpPCA function ("Kfold method" for 166 cross-validation); imputation of the data set with the impute”. However, this procedure refers to the estimation of missing values. Authors must mention in the Materials and Methods and in the Results if missing values were in the original data file and at which extent.

Authors again show results of a Principal Component Analysis (PCA) to test for effects of soil parameters on the growth of Trichoderma species under 2 different soil treatments (AS and NS).
However, in this experiment Authors have TWO different groups of variables, one related to soil parameters and one group related to Trichoderma growth.

PCA can be applied when ONE SET of variables is under study.

Authors could apply PCA to evaluate soil differences among the 13 locations where soil samples were collected (for example using the data shown in Table 2).

Authors could also apply PCA to evaluate the growth of Trichoderma on soils sampled in the 13 locations (here again E2 must be removed because of missing treatments AST029 and AST059).

However, in this way the relationship between soil parameters and Trichoderma growth cannot be established.

If Authors want to relate soil variables to Trichoderma growth, they must use other multivariate approaches such as Canonical Correlation Analysis. This approach will produce pairs of linear functions that allow to establish the relative relationship between the variables of the first group (Soil parameters) and of the second group (Trichoderma growth).

I suggest Authors to re-evaluate the multivariate approach applied to find a relationship between soil parameters and Trichoderma growth.

Author Response

ANOVA

Authors excluded the E2 location and carried out an ANOVA2.

The Table 1 and the description of sites where soil samples were collected must be in the Materials and Methods section, not in the results.

The Table 1 was put in Material and Methods. Section 2.3 Soil sampling.

However, as shown in Figure 2, the ANOVA has 3 main factors: Trichoderma species (T029 and T059), Soil Treatment (Autoclaved vs Natural) and Locations (from A1 to P5). Therefore, it is a 3 way ANOVA and Figure 2 shows the second order interaction (Species x Soil Treatment x Location). Therefore, in Materials and Methods the Authors should say 3way-ANOVA. Moreover, an ANOVA Table must be added as supplementary table to evidence the significance of variances for each source of variation: main effects and interactions.

We have made the analysis Three-way ANOVA and we have added the results in the Supplementary material.

PCA

Authors did not mention the standardization of data before performing PCA, that is necessary if the different variables show very different variances. Moreover, Authors refer to the use of R by  “estim_ncpPCA function ("Kfold method" for 166 cross-validation); imputation of the data set with the impute”. However, this procedure refers to the estimation of missing values. Authors must mention in the Materials and Methods and in the Results if missing values were in the original data file and at which extent.

You are right, Kfold method was used previously and it was not removed. Now, there is not missing values so we will not use it.

Authors again show results of a Principal Component Analysis (PCA) to test for effects of soil parameters on the growth of Trichoderma species under 2 different soil treatments (AS and NS).

However, in this experiment Authors have TWO different groups of variables, one related to soil parameters and one group related to Trichoderma growth. PCA can be applied when ONE SET of variables is under study. Authors could apply PCA to evaluate soil differences among the 13 locations where soil samples were collected (for example using the data shown in Table 2).

Authors could also apply PCA to evaluate the growth of Trichoderma on soils sampled in the 13 locations (here again E2 must be removed because of missing treatments AST029 and AST059).

However, in this way the relationship between soil parameters and Trichoderma growth cannot be established. If Authors want to relate soil variables to Trichoderma growth, they must use other multivariate approaches such as Canonical Correlation Analysis. This approach will produce pairs of linear functions that allow to establish the relative relationship between the variables of the first group (Soil parameters) and of the second group (Trichoderma growth).

I suggest Authors to re-evaluate the multivariate approach applied to find a relationship between soil parameters and Trichoderma growth.

Following your instructions, we have performed a Canonical Correlation Analysis for doing this correlation. Two different datasets were considered (Trichoderma and soil data) and their relationships were described in the paper. Even, soil data were divided in nutrients and soil parameters.

We have added these results in the lines 272-309. Also. We have made some figures by test and the supplementary material (Figures S1-S3).
